# SUPPORT IS ALL YOU NEED FOR CERTIFIED VAE TRAINING

**Changming Xu, Debangshu Banerjee, Deepak Vasisht & Gagandeep Singh**
Department of Computer Science
University of Illinois Urbana-Champaign
Champaign, IL 61820, USA
{cx23, db21, deepakv, ggnds}@illinois.edu

## ABSTRACT

Variational Autoencoders (VAEs) have become increasingly popular and deployed in safety-critical applications. In such applications, we want to give certified probabilistic guarantees on performance under adversarial attacks. We propose a novel method, CIVET, for certified training of VAEs. CIVET depends on the key insight that we can bound worst-case VAE error by bounding the error on carefully chosen support sets at the latent layer. We show this point mathematically and present a novel training algorithm utilizing this insight. We show in an extensive evaluation across different datasets (in both the wireless and vision application areas), architectures, and perturbation magnitudes that our method outperforms SOTA methods achieving good standard performance with strong robustness guarantees.

## 1 INTRODUCTION

Deep neural networks (DNNs) achieve state-of-the-art performance in a wide range of fields, including wireless communications Cho et al. (2023); Yang et al. (2018), autonomous driving Bojarski et al. (2016); Shafaei et al. (2018), and medical diagnosis Amato et al. (2013); Kononenko (2001). Despite their success, DNNs are vulnerable to adversarial perturbations added to the input, making their use in safety-critical systems, such as autonomous driving and wireless communication, risky and potentially life-threatening. To address this issue, numerous robust learning Mirman et al. (2018); Mao et al. (2023); Wong & Kolter (2018) and verification approaches Singh et al. (2019); Xu et al. (2021); Wang et al. (2018); Ehlers (2017) have been developed for deterministic DNNs. However, robust learning approaches for stochastic DNNs, including popular generative deep neural networks, are scarce. With the recent surge in the use of stochastic networks like variational autoencoders (VAEs) in security-critical systems, such as wireless networks Liu et al. (2021), it is increasingly vital to develop training methods for stochastic DNNs which are accurate and have provable guarantees of robustness.

A VAE is a generative deep neural network architecture. VAEs are used in various domains such as computer vision Duan et al. (2023), language processing Qian & Cheung (2019), wireless Liu et al. (2021), and representation learning van den Oord et al. (2017). However, as with other neural network architectures, existing research has shown that VAE's performance can be unreliable when exposed to adversarial attacks Kos et al. (2018); Gondim-Ribeiro et al. (2018). The few existing works on training VAEs with formal guarantees impose strict architectural constraints like fixed latent layer variance Barrett et al. (2022). We lift these restrictions and propose a general framework for training VAEs with certified robustness.

**Key Challenges**. Unlike deterministic DNN classifiers commonly used in certifiably robust training methods Gowal et al. (2018); Yang et al. (2023); Mueller et al. (2022), VAE's outputs are stochastic. This requires training methods that can compute the worst-case loss over a potentially infinite set of output distributions. Even for a single output distribution, calculating the worst-case error is challenging, as its probability density function may not have a tractable closed form. Additionally, to apply these methods to practical networks, the worst-case error computation must be both fast and compatible with gradient descent based optimization methods, which are typically used in DNN

training. Therefore, the worst-case error must be expressed as a differentiable program involving network parameters to enable efficient parameter refinement.

**This work**. We propose Certified Interval Variational Autoencoder Training (CIVET), which efficiently bounds the worst-case performance of VAEs over an input region while ensuring that the error bound remains differentiable and useful for optimizing network parameters. To the best of our knowledge, CIVET is the first certified training algorithm for VAEs that imposes no Lipschitz or fixed variance constraints on the architecture. Our method is based on a *key insight*: by carefully selecting a subset (support set) of the latent space and bounding the worst-case error of the *deterministic* decoder within this set, we can effectively bound the worst-case error across all reachable output distributions. Here, the support set selection is driven by the distributions computed at the latent layer by the encoder.

**Main Contributions**. We list our main contributions below:

- We mathematically show that for VAEs, the worst-case error over all reachable output distributions from an input region can be bounded in two steps: (a) identifying an appropriate subset $\mathcal{S}$ of the latent space (the support set), and (b) bounding the worst-case error of the decoder over $\mathcal{S}$. This reduction simplifies the problem of bounding the worst-case error for stochastic VAEs to a more tractable problem of bounding the error for deterministic decoder networks. However, both steps - support set selection and decoder error bounding must be differentiable to enable efficient learning.
- By restricting the support sets to specific geometric shapes, such as multidimensional boxes, we ensure that both the support selection and bounding steps are differentiable. Here, the support selection step relies on the encoder parameters, while the error bounding step involves the decoder parameters. CIVET efficiently combines these two steps, enabling the simultaneous optimization of the encoder and decoder parameters to minimize the worst-case error w.r.t. the input region.
- We perform extensive experiments across the wireless and vision domains on popular datasets with different DNN architectures, showing that our method significantly improves robust worst-case errors while causing only a small degradation in standard non-adversarial settings[1].

## 2 BACKGROUND

This section provides the necessary notations, definitions, and background on deterministic and stochastic DNN certification, and certified robust training methods for deterministic DNNs.

**Notation**. Throughout the rest of the paper, we use small case letters $(x, y)$ for constants, bold small case letters $(\boldsymbol{x}, \boldsymbol{y})$ for vectors, capital letters $X, Y$ for functions and random variables, and calligraphed capital letters $\mathcal{X}, \mathcal{Y}$ for sets including sets of probability distributions.

### 2.1 VARIATIONAL AUTOENCODERS

Given a set of inputs, $X \subseteq \mathbb{R}^{d_{\text{in}}}$, generated via an unknown process with latent variables, $\mathbf{Z} \subseteq \mathbb{R}^{d_{\text{l}}}$, we want to learn a latent variable model with joint density $p_\theta(\mathbf{x}, \mathbf{z}) = p_\theta(\mathbf{x}|\mathbf{z})p(z)$ which describes this process. Learning the parameterization, $\theta$, is often intractable via maximum likelihood; instead, we can use variational inference to address this intractability by learning a conditional likelihood model $p_{\theta_d}(x|z)$ and an approximated posterior distribution $p_{\theta_e}(z|x)$ Kingma et al. (2019). A Variational Autoencoder (VAE) is a combination of these two models where $\theta_e$ represents the parameters of an encoder network, $N^e : \mathbb{R}^{d_{\text{in}}} \to \mathcal{P}(\mathbb{R}^{d_{\text{l}}})$, and $\theta_d$ represents the parameters of a decoder network, $N^d : \mathcal{P}(\mathbb{R}^{d_{\text{l}}}) \to \mathcal{P}(\mathbb{R}^{d_{\text{out}}})$. Here $\mathcal{P}(\mathbb{R}^n)$ denotes the set of probability distributions defined over $\mathbb{R}^n$. Generally, for VAEs, given a single input $\boldsymbol{z} \in \mathbb{R}^{d_{\text{l}}}$, the decoder's output $N^d(\boldsymbol{z})$ is deterministic.

**VAEs in Wireless**. The effectiveness of VAEs has resulted in their use in several security-critical systems, including wireless applications Liu et al. (2021); Cho et al. (2023); Yang et al. (2018). Liu et al. (2021) present FIRE, an end-to-end machine learning approach that utilizes VAEs for precise channel estimation, a critical operation for wireless communications such as cellular and Wi-Fi networks. In a real-world testbed environment, FIRE achieves an SNR (Signal to Noise Ratio) improvement of over 10 dB in Multiple Input Multiple Output (MIMO) transmissions compared to SOTA non-machine learning methods. However, despite their strong performance in wireless systems, Liu et al. (2023) exposes the vulnerabilities of VAEs to practical adversarial attacks.

---

[1]Code is provided at https://github.com/uiuc-focal-lab/civet

## 2.2 Neural Network Certification

Given a deterministic DNN $N : \mathbb{R}^{d_{\text{in}}} \to \mathbb{R}^{d_{\text{out}}}$, DNN certification Singh et al. (2019) proves that the network outputs $\boldsymbol{y} = N(\boldsymbol{x})$ corresponding to all possible inputs $\boldsymbol{x}$ specified by input specification $\phi : \mathbb{R}^{d_{\text{in}}} \to \{True, False\}$, satisfy output specification $\psi : \mathbb{R}^{d_{\text{out}}} \to \{True, False\}$. Formally, we show that $\forall \boldsymbol{x} \in \mathbb{R}^{d_{\text{in}}}.\phi(\boldsymbol{x}) \implies \psi(N(\boldsymbol{x}))$ holds. Safety properties like local DNN robustness encode the output specification ($\psi$) as a linear inequality (or conjunction of linear inequalities) over DNN output $\boldsymbol{y} \in \mathbb{R}^{d_{\text{out}}}$. e.g. $\psi(\boldsymbol{y}) = (\boldsymbol{c}^T \boldsymbol{y} \geq 0)$ where $\boldsymbol{c} \in \mathbb{R}^{d_{\text{out}}}$. However, this formulation assumes $N$ is deterministic and does not work when the network's output is stochastic like VAEs.

**Certification for Stochastic Networks**. (Berrada et al., 2021) generalizes DNN certification for stochastic networks including VAEs w.r.t. the input region $\phi_t \subseteq \mathbb{R}^{d_{\text{in}}}$. At a high level, in this case, we want to prove that the worst-case error $\epsilon$ is bounded over all possible output distributions with high probability say $(1 - \delta) \in (0.5, 1]$. For a set of inputs (possibly infinite) $\phi_t \subseteq \mathbb{R}^{d_{\text{in}}}$, let $\mathcal{Y}$ denote the set of distributions computed by a stochastic DNN $N : \mathbb{R}^{d_{\text{in}}} \to \mathcal{P}(\mathbb{R}^{d_{\text{out}}})$ on $\phi_t$. Then $\mathcal{Y} = \{Y | Y = N(\boldsymbol{x}), \boldsymbol{x} \in \mathbb{R}^{d_{\text{in}}}\}$ where $Y$ is a random variable representing the output distribution $N(\boldsymbol{x})$ at a single input $\boldsymbol{x}$. Given an error threshold $\epsilon_0 \in \mathbb{R}$, target probability threshold $(1 - \delta) \in (0.5, 1]$, and a error function $M : \mathbb{R}^{d_{\text{out}}} \to [0, \infty)$ the output specification $\psi : \mathcal{P}(\mathbb{R}^{d_{\text{out}}}) \to \{True, False\}$ is defined as $P_{min}(\mathcal{Y}, \epsilon_0) \geq (1 - \delta)$ with $P_{min}(\mathcal{Y}, \epsilon_0) = \min_{Y \in \mathcal{Y}} P(M(Y) \leq \epsilon_0)$.

## 2.3 Certified training for deterministic DNNs

Certified training allows to learn network parameters that make the DNN provably robust against adversarial perturbations. During training for any input $\boldsymbol{x_0} \in \mathbb{R}^{d_{\text{in}}}$ from the training set, certified training methods first bound the worst-case loss $\mathcal{L}_w(N_\theta(\boldsymbol{x_0})) = \max_{\boldsymbol{x} \in \phi_t(\boldsymbol{x_0})} \mathcal{L}(N_\theta(\boldsymbol{x}))$ w.r.t. a local input region $\phi_t(\boldsymbol{x_0})$ around $\boldsymbol{x_0}$. The method then refines the network parameters $\theta$ based on the worst-case loss $\mathcal{L}_w(N_\theta(\boldsymbol{x_0})$ instead of the point-wise loss $\mathcal{L}(N_\theta(\boldsymbol{x_0}))$ used in standard training. The local input region $\phi_t(\boldsymbol{x_0})$ typically contains all possible inputs $\boldsymbol{x}$ satisfying $\|\boldsymbol{x} - \boldsymbol{x_0}\|_\infty \leq \epsilon$ for some perturbation budget $\epsilon \in \mathbb{R}^+$. However, even for $ReLU$ networks and $L_\infty$ norm bounded local input regions, exactly computing $\mathcal{L}_w(N_\theta(\boldsymbol{x_0}))$ is NP-Hard Katz et al. (2017b). Hence, SOTA certified training methods for scalability replace the worst-case loss $\mathcal{L}_w(N_\theta(\boldsymbol{x_0}))$ with an efficiently computable upper bound $\mathcal{L}_w(N_\theta(\boldsymbol{x_0})) \leq \mathcal{L}_{ub}(N_\theta(\boldsymbol{x_0}))$. Note that minimizing the *upper bound* $\mathcal{L}_{ub}(N_\theta(\boldsymbol{x_0}))$ provably reduces the worst-case loss during optimization. Moreover, training methods ensure that $\mathcal{L}_{ub}(N_\theta(\boldsymbol{x_0}))$ computation can be expressed as a differentiable program allowing to refine network parameters $\theta$ with gradient descent based algorithms. For example, one popular certified training method Mirman et al. (2018) uses interval bound propagation (IBP) or Box propagation to compute $\mathcal{L}_{ub}(N_\theta(\boldsymbol{x_0}))$. IBP first over-approximates the input region, $\phi_t(\boldsymbol{x_0})$, with a multidimensional box where each dimension $i \in [d_{\text{in}}]$ is an interval with bounds $[l_i, u_i]$. Then IBP training propagates the input box through each layer of the network using interval arithmetic Mirman et al. (2018) to find the bounding interval $[\mathcal{L}_{lb}(N_\theta(\boldsymbol{x_0})), \mathcal{L}_{ub}(N_\theta(\boldsymbol{x_0}))]$ of the worst-case error $\mathcal{L}_w(N_\theta(\boldsymbol{x_0}))$.

Although significant progress has been made in certifiable robust training for deterministic networks, to the best of our knowledge, there is currently no general framework for certifiably robust training of VAEs that does not impose architectural constraints. The key challenge is to bound the worst-case training loss over a potentially infinite set of output probability distributions while ensuring that the bounding method is both scalable and differentiable (suitable for gradient-based parameter learning). Next, we discuss our approach for training certifiably robust VAEs.

## 3 Certifiably Robust Training for VAEs

In this section, we describe the key steps of CIVET with the formal problem formulation in Sec. 3.1, the reduction of worst-case error computation for stochastic VAEs to worst-case error bounding for deterministic decoders in Sec. 3.2 and efficient support selection and bounding algorithm in Sec. 3.3. Fig 1 illustrates the workings of CIVET highlighting its key steps. In Appendix C we provide a concrete example of the steps outlined in this section.

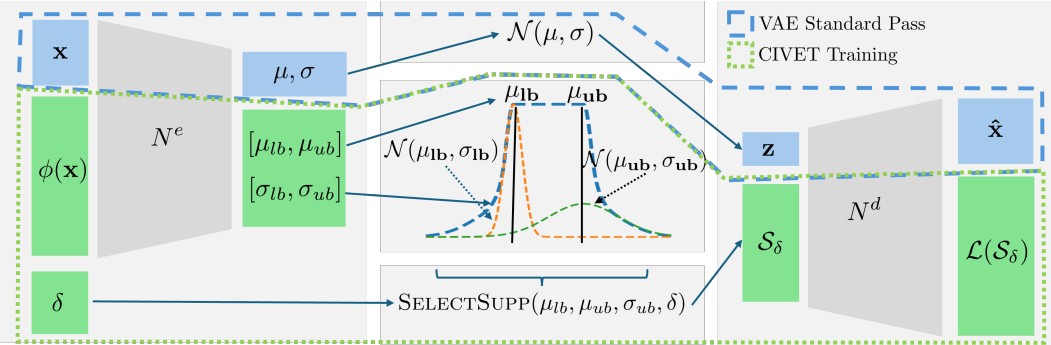

Figure 1: (CIVET Overview) The blue dashed box ( ) shows a standard pass over a VAE, where the encoder ($N^e$) generates a parameterization of a distribution which is sampled in the latent layer and passed to the deterministic decoder ($N^d$). The green dotted box ( ) shows CIVET training over the same VAE. Here an input region is passed through $N^e$ using a deterministic DNN bounding algorithm like IBP which gives a range of distribution parameterizations. CIVET then computes a support set with a given probability threshold $(1 - \delta)$ which can then be passed through $N^d$ using a deterministic DNN bounding algorithm to obtain an overapproximation of the loss.

## 3.1 PROBLEM FORMULATION

We define the worst-case loss of a VAE with an encoder network $N^e$ and decoder network $N^d$ w.r.t. an input region $\phi_t(\boldsymbol{x_0})$ around a training data point $\boldsymbol{x_0} \in \mathbb{R}^{d_{\text{in}}}$. Let, $\mathcal{Z}$ denote the set of distributions at the latent layer computed by $N^e$ on $\phi_t(\boldsymbol{x_0})$ i.e. $\mathcal{Z} = \{Z \mid Z = N^e(\boldsymbol{x}), \boldsymbol{x} \in \phi_t(\boldsymbol{x_0})\}$ and $\mathcal{Y}$ be the set of output distributions $\mathcal{Y} = \{Y \mid Y = N^d(Z), Z \in \mathcal{Z}\}$. Note that each $Y$ and $Z$ are random variables corresponding to a specific probability distribution over $\mathbb{R}^{d_l}$ and $\mathbb{R}^{d_{\text{out}}}$ respectively. Then given a target probability threshold $(1 - \delta)$ and error function $M$, the worst case error $\mathcal{L}_w(N^e, N^d, \boldsymbol{x_0}) = \max_{Y \in \mathcal{Y}} T(Y)$ where $T(Y)$ is defined as follows

$$T(Y) = \min_{\omega \in \mathbb{R}} \omega \quad \text{s.t} \quad P(M(Y) \leq \omega) \geq (1 - \delta) \tag{1}$$

At a high level, for any given output distribution $Y$, $T(Y)$ determines the tightest possible error threshold, ensuring that for any sample $\boldsymbol{y} \sim Y$, the corresponding error $M(\boldsymbol{y})$ is no more than $T(Y)$ with a probability of at least $(1 - \delta)$. $\mathcal{L}_w(N^e, N^d, \boldsymbol{x_0})$ maximizes the error threshold $T(Y)$ over all possible output distributions. Assuming $\boldsymbol{x}$ sampled from input distribution $X$ the expected worst case loss is $\mathbb{E}_{\boldsymbol{x} \sim X} \mathcal{L}_w(N^e, N^d, \boldsymbol{x})$. Now, with fixed architectures of $N^e$ and $N^d$ learning the parameters $\theta_e, \theta_d$ corresponding to the smallest expected worst case loss can be reduced to the following optimization problem: $(\theta_e^*, \theta_d^*) = \arg\min_{\theta_e, \theta_d} \mathbb{E}_{\boldsymbol{x} \sim X}.\mathcal{L}_w(\theta_e, \theta_d, \boldsymbol{x})$. While this optimization problem precisely defines the optimal parameters $(\theta_e^*, \theta_d^*)$, solving it exactly is computationally prohibitive for networks of practical size. Even determining $T(Y)$ for a single continuous random variable $Y$ can be costly, making worst-case loss computation for a single local input region $\phi_t(\boldsymbol{x_0})$ practically intractable. Therefore, similar to certifiably robust training methods for deterministic networks, we focus on computing a mathematically sound upper bound $\mathcal{L}_{ub}(\theta_e, \theta_d, \boldsymbol{x_0})$.

However, unlike deterministic DNNs that existing works handle, VAEs require bounding methods capable of handling a potentially infinite set of probability distributions. To tackle this, we first show it is possible to compute a non-trivial upper bound $\mathcal{L}_{ub}(\theta_e, \theta_d, \boldsymbol{x_0})$ by: a) finding an appropriate subset $\mathcal{S} \subseteq \mathbb{R}^{d_l}$ (referred as the support set in the rest of the paper) based on the set of reachable distributions in the latent layer, and b) bounding the worst case error of the decoder $N^d$ over $\mathcal{S}$ i.e. bounding $\max_{\boldsymbol{z} \in \mathcal{S}} M(N^d(\boldsymbol{z}))$ (Section 3.2). Additionally, in Section 3.3, we show that both finding and bounding $\mathcal{S}$ can be efficiently expressed as a differentiable program involving the encoder and decoder parameters $\theta_e, \theta_d$. This enables learning the parameters with gradient descent.

## 3.2 BOUNDING WORST-CASE LOSS

First, we formally define support sets for the set of distributions $\mathcal{Z}$ at the latent layer.

**Definition 1** (Support Sets). *For a set of distributions $\mathcal{Z}$ over $\mathbb{R}^{d_l}$ and a probability threshold $(1-\delta)$, a subset $\mathcal{S} \subseteq \mathbb{R}^{d_l}$ is a support for $\mathcal{Z}$ provided $\big( \min_{Z \in \mathcal{Z}} P(Z \in \mathcal{S}) \big) \geq (1 - \delta)$.*

For fixed $(1 - \delta)$, we show that for **any** support set $\mathcal{S}$ the error upper bound $T_{ub}(\mathcal{S}) = \max_{\boldsymbol{z} \in \mathcal{S}} M(N^d(\boldsymbol{z}))$ serves as valid upper bound of the worst-case error $\mathcal{L}_w(\theta_e, \theta_d, \boldsymbol{x_0}) \leq T_{ub}(\mathcal{S})$ (Thm 1). Since decoder's output $N^d(\boldsymbol{z})$ is deterministic for all $\boldsymbol{z} \in \mathcal{S}$, computing $T_{ub}(\mathcal{S})$ is same as bounding the error of a deterministic ($N^d$) network on input region ($\mathcal{S}$). This shows that with appropriate $\mathcal{S}$ we can reduce the worst-case error bounding for VAEs to the worst-case error bounding of deterministic networks as handled in existing works Katz et al. (2017b).

**Theorem 1.** *For a VAE with encoder $N^e$, decoder $N^d$, local input region $\phi_t(\boldsymbol{x_0})$, error function $M$ and probability threshold $(1 - \delta)$, if $\mathcal{Z} = \{Z \mid Z = N^e(\boldsymbol{x}), \boldsymbol{x} \in \phi_t(\boldsymbol{x_0})\}$ then for any support set $\mathcal{S}$ for $\mathcal{Z}$ the worst case error $\mathcal{L}_w(N^e, N^d, \boldsymbol{x_0}) \leq T_{ub}(\mathcal{S})$ where $T_{ub}(\mathcal{S}) = \max_{z \in \mathcal{S}} M(N^d(\boldsymbol{z}))$.*

**Proof Sketch**. Let, $I_{t_0}(\boldsymbol{y}) = (M(\boldsymbol{y}) \leq t_0)$ be an indicator where $t_0 = T_{ub}(\mathcal{S})$. The key observations are that - a) the indicator $I_{t_0}(N^d(\boldsymbol{z})) = 1$ for all $\boldsymbol{z} \in \mathcal{S}$ and b) given $\mathcal{S} \subseteq \mathbb{R}^{d_l}$ is a support hence for any $Z \in \mathcal{Z}$, $\int_{\mathcal{S}} f_Z(\boldsymbol{z}) d\boldsymbol{z} \geq (1 - \delta)$ where $f_Z$ is the probability density function of $Z$. Now, with Eq. 1, it is enough to show $P(M(Y) \leq t_0) \geq (1 - \delta)$ for any $Y = N^d(Z)$ where $Z \in \mathcal{Z}$. Since $\mathcal{S} \subseteq \mathbb{R}^{d_l}$, $P(M(Y) \leq t_0) \geq \int_{\mathcal{S}} I_{t_0}(N^d(\boldsymbol{z})) \times f_Z(\boldsymbol{z}) d\boldsymbol{z}$ and from observations (a) and (b) we get $\int_{\mathcal{S}} I_{t_0}(N^d(\boldsymbol{z})) \times f_Z(\boldsymbol{z}) d\boldsymbol{z} \geq (1 - \delta)$. The detailed proof is in Appendix A.

Ideally from the set of all possible supports $\mathbb{S}$, we should pick the support $\mathcal{S}^*$ that minimizes $T_{ub}(\mathcal{S})$ over all $\mathcal{S} \in \mathbb{S}$. Although $\mathcal{S}^*$ provides the tightest upper bound on $\mathcal{L}_w(N^e, N^d, \boldsymbol{x_0})$ from $\mathbb{S}$, finding and subsequently bounding the optimal $\mathcal{S}^*$ can be expensive. In contrast, picking an arbitrary support $\mathcal{S}$ from $\mathbb{S}$ can make the upper bound on $\mathcal{L}_w(N^e, N^d, \boldsymbol{x_0})$ too loose hurting parameter $(\theta_e, \theta_d)$ refinement. For example, the entire latent space $\mathbb{R}^{d_l}$ is always a valid support but it fails to provide a non-trivial upper bound on $\mathcal{L}_w(N^e, N^d, \boldsymbol{x_0})$. To strike a balance between computational efficiency and tightness of the computed bounds we first restrict $\mathbb{S}$ to a subset of supports $\mathbb{S}' \subseteq \mathbb{S}$ where computing $T_{ub}(\mathcal{S})$ is cheap and then find the optimal support $\mathcal{S}^*$ from this restricted set (Section 3.3).

Before moving into the support selection algorithm, we want to emphasize a couple of key points. First, Theorem 1 can be extended to subsets of any hidden decoder layer. This means that supports can be chosen from any hidden decoder layer, not just the latent layer. However, we focus on the latent layer because the distributions $Z \in \mathcal{Z}$ typically have well-defined closed forms for their probability density functions, such as Gaussian distributions, which simplifies the support selection process. Second, similar to stochastic DNN verifiers Berrada et al. (2021); Wicker et al. (2020), we assume that for a given $(1 - \delta)$, there exists at least one bounded support $\mathcal{S}$. Without this, $T_{ub}(\mathcal{S})$ for an arbitrary unbounded $\mathcal{S}$ may not be bounded.

### 3.3 SUPPORT SET COMPUTATION AND BOUNDING

As mentioned above, it is computationally expensive to minimize over all possible supports $\mathbb{S}$. By picking a restricted set of supports $\mathbb{S}'$ we balance computational efficiency and tightness. In $\mathbb{S}'$, we only consider multi-dimensional boxes where for any dimension $i \in [d_l]$ we include all values within the range $[l_i, u_i]$. Hence, finding a support set from $\mathbb{S}'$ only requires computing the bounds $l_i, u_i$ for each dimension. Moreover, IBP techniques commonly used in deterministic certified training can bound the worst-case error of the decoder over any support set from $\mathbb{S}'$. So only picking a support set is sufficient to reduce the problem into an instance of deterministic certified training.

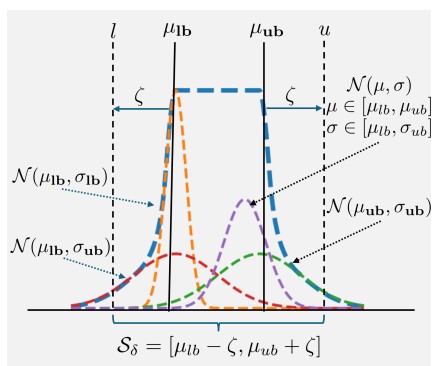

Figure 2: Support Set Visualization. Given a set of distributions $\{\mathcal{N}(\mu, \sigma) | \mu \in [\mu_{lb}, \mu_{ub}], \sigma \in [\sigma_{lb}, \sigma_{ub}]\}$ we define a symmetric support set $S_\delta = [\mu_{lb} - \zeta, \mu_{ub} + \zeta]$

Given a set of distributions $Z \in \mathcal{Z}$ with probability density functions $f_Z(\boldsymbol{z})$ and a fixed $(1 - \delta)$, let $S = [\boldsymbol{l}, \boldsymbol{u}]$. Our bounding problem can now be expressed as finding $\boldsymbol{l}, \boldsymbol{u}$ s.t. $\forall Z \in \mathcal{Z}. \int_{\boldsymbol{l}}^{\boldsymbol{u}} f_Z(\boldsymbol{z}) d\boldsymbol{z} \geq (1 - \delta)$.

For the scope of this paper, we assume that $\mathcal{Z}$ is derived from the latent layer of a VAE where $N^e$ is a deterministic network that outputs $\boldsymbol{\mu}, \boldsymbol{\sigma}$ for each latent dimension parameterizing a gaussian distribution (the most common setting for VAEs). Thus, given $\phi_t(\boldsymbol{x_0})$ and $N^e$ we can use existing deterministic network bounding techniques (i.e. Mirman et al. (2018)) to overapproximate reachable

intervals $[\boldsymbol{\mu_{lb}}, \boldsymbol{\mu_{ub}}], [\boldsymbol{\sigma_{lb}}, \boldsymbol{\sigma_{ub}}]$. Let's first consider the 1d case. We now have $\mathcal{Z} = \{\mathcal{N}(\mu, \sigma) \mid \mu \in [\mu_{lb}, \mu_{ub}], \sigma \in [\sigma_{lb}, \sigma_{ub}]\}$. Therefore, our bounding problem can be expressed as finding $l, u$ s.t.

$$\forall \mu \in [\mu_{lb}, \mu_{ub}], \sigma \in [\sigma_{lb}, \sigma_{ub}].\Phi_{\mu,\sigma}(u) - \Phi_{\mu,\sigma}(l) \geq (1 - \delta) \tag{2}$$

Here, $\Phi_{\mu,\sigma}$ is the normal cumulative distribution function (CDF) with mean $\mu$ and variance $\sigma^2$, and for ease of notation let $\Phi := \Phi_{0,1}$. There are many ways to pick $[l, u]$ satisfying Equation 2. Since this set of normal distributions is symmetric around its midpoint $1/2(\mu_{lb} + \mu_{ub})$, we choose a symmetric support, i.e. $[l, u] = [\mu_{lb} - \zeta, \mu_{ub} + \zeta]$ with $\zeta > 0$. Note that the interval $[l, u]$ always includes the interval $[\mu_{lb}, \mu_{ub}]$ as required by the condition $(1 - \delta) > 0.5$ (see Sec. 2.2). Figure 2 gives a pictorial representation of our support selection. In Theorem 2, given $(1 - \delta)$ we give a support set satisfying Equation 2.

**Lemma 1.** *Given bounds $\mu_{lb}, \mu_{ub}, \sigma_{lb}, \sigma_{ub} \in \mathbb{R}$, probability threshold $(1 - \delta)$, and $\zeta \in \mathbb{R}^+$. Let $\mathcal{C}(\mu, \sigma, u, l) = \Phi_{\mu,\sigma}(u) - \Phi_{\mu,\sigma}(l)$. Then $\forall \mu \in [\mu_{lb}, \mu_{ub}], \sigma \in [\sigma_{lb}, \sigma_{ub}]$,*

$$\mathcal{C}(\mu, \sigma, \mu_{ub} + \zeta, \mu_{lb} - \zeta) \geq \mathcal{C}(\mu_{ub}, \sigma_{ub}, \mu_{ub} + \zeta, \mu_{lb} - \zeta)$$

*Furthermore,*

$$\mathcal{C}(\mu_{lb}, \sigma_{ub}, \mu_{ub} + \zeta, \mu_{lb} - \zeta) = \mathcal{C}(\mu_{ub}, \sigma_{ub}, \mu_{ub} + \zeta, \mu_{lb} - \zeta)$$

**Proof Sketch**. The proof relies on two key facts: a) for a fixed mean $\mu \in [\mu_{lb}, \mu_{ub}]$ and bounds $l \leq \mu_{lb}$, $\mu_{ub} \leq u$, $\sigma < \sigma_{ub} \implies \mathcal{C}(\mu, \sigma, u, l) > \mathcal{C}(\mu, \sigma_{ub}, u, l)$ b) for any standard deviation $\sigma \in [\sigma_{lb}, \sigma_{ub}]$ $\mathcal{C}(\mu, \sigma, \mu_{ub} + \zeta, \mu_{lb} - \zeta)$ is maximized at $\mu = \frac{\mu_{lb} + \mu_{ub}}{2}$ and is strictly decreasing on each side as $\mu$ increases (or, decreases). Formal proof is provided in Appendix A.

Lemma 1 implies that when a support set is symmetric around $[\mu_{lb}, \mu_{ub}]$ then the distributions at the endpoints $\mathcal{N}(\mu_{lb}, \sigma_{ub}), \mathcal{N}(\mu_{lb}, \sigma_{ub})$ minimize $\mathcal{C}$.

**Theorem 2.** *Given $\mathcal{Z} = \{\mathcal{N}(\mu, \sigma) \mid \mu \in [\mu_{lb}, \mu_{ub}], \sigma \in [\sigma_{lb}, \sigma_{ub}]\}$, probability threshold $(1 - \delta)$, then $[l, u] = [\mu_{lb} - \sigma_{ub}\Phi^{-1}(p_0), \mu_{ub} + \sigma_{ub}\Phi^{-1}(p_0)]$ satisfies Equation 2. Where*

$$p_0 = \min_{p \in [(1-\delta),1]} \left[ \Phi^{-1}(p) + \Phi^{-1}(p - (1 - \delta)) \geq \frac{\mu_{lb} - \mu_{ub}}{\sigma_{ub}} \right]$$

.

**Proof Sketch**. Using Lemma 1, we can prove that this formulation satisfies Equation 2 if $\mathcal{N}(\mu_{ub}, \sigma_{ub})$ does. We can then plug this distribution into $\mathcal{C}$ to show that $\Phi_{\mu_{ub}, \sigma_{ub}}(u) - \Phi_{\mu_{ub}, \sigma_{ub}}(l) \geq (1 - \delta)$. Formal proof can be found in Appendix A.

Although $\Phi^{-1}$ has no closed form solution Vedder (1993), given $\boldsymbol{\mu_{lb}}, \boldsymbol{\mu_{ub}}, \boldsymbol{\sigma_{ub}}$ we can compute $p_0$ using binary search since $\Phi^{-1}$ is strictly monotonic. For higher dimensional latent spaces, the distribution for each dimension in the latent layer is independent for individual inputs. Based on this, given a probability threshold $(1 - \delta)$ over $d_l$ dimensions, we can compute the probability threshold for each dimension as $(1 - \delta)^{1/d_l}$. Therefore, given $(1 - \delta)$, each dimension $i \in [d_l]$ of the selected support $\mathcal{S}_\delta$ is an interval $\mathcal{S}_\delta^i = [\mu_{lb}^i - \sigma_{ub}^i\Phi^{-1}(p_0^i), \mu_{ub}^i + \sigma_{ub}^i\Phi^{-1}(p_0^i)]$ where $p_0^i$ is independently computed in each dimension as defined in Theorem 2. This shows the $\mathcal{S}_\delta$ is a function of the encoder's output $(\boldsymbol{\mu_{lb}}, \boldsymbol{\mu_{ub}}, \boldsymbol{\sigma_{ub}})$ and subsequently depends on the encoder parameters $\theta_e$.

## 3.4 CIVET LOSS

Theorem 2 gives us a way to find a support set, $\mathcal{S}_\delta$ given $\boldsymbol{\mu_{lb}}, \boldsymbol{\mu_{ub}}, \boldsymbol{\sigma_{ub}}$ for a specific $(1 - \delta)$. However, we may not know the target probability at training time. We would like to cover the target probability without selecting an overly large $(1 - \delta)$ which may result in a loose bound. Inspired by numerical integration Gibb (1915), CIVET computes the loss over multiple support sets weighting them based on their covered probability. We provide additional analysis on this selection in Appendix D.

Given a single $\delta$, we can obtain an overapproximation of $T_{ub}(\mathcal{S}_\delta)$ by passing $\mathcal{S}_\delta$ through the deterministic ($N^d$) network using existing worst-case network bounding techniques Mirman et al. (2018). We call this loss $\mathcal{L}_{dec}(N^d, \boldsymbol{x}, \mathcal{S}_\delta)$. For CIVET we combine multiple support sets to compute our loss so we let $\mathcal{S}_{\delta_i}$ be the support for $\mathcal{Z}$ with probability threshold $(1 - \delta_i)$ for each $i$. For the remainder of the paper, assume that lists of $\delta$ are sorted in reverse order, formally, $\delta_i < \delta_j$ if $i > j$. For the largest $\delta$, $\delta_1$, corresponding to the smallest $\mathcal{S}_{\delta_1}$ we assign weight $(1 - \delta_1)$. For the remainder of the $\delta_i$'s, we weigh them based on the additional probability they cover compared to the previous $\delta$, in other words, $\forall i \in [2, \ldots, n].\mathcal{S}_{\delta_i}$ gets weight $\delta_{i-1} - \delta_i$. This leads to CIVET loss.

**Definition 2.** *(CIVET Loss) Given a deterministic decoder network $N^d$, input $\boldsymbol{x} \in \mathbb{R}^{d_{in}}$, $\boldsymbol{\mu_{lb}}, \boldsymbol{\mu_{ub}}, \boldsymbol{\sigma_{ub}} \in {}_l$, and a set of $\delta s$ $\{\delta_1, \ldots, \delta_n\}$. We define,*

$$\mathcal{L}_{CIVET} = (1 - \delta_1)\mathcal{L}_{dec}(N^d, \boldsymbol{x}, \mathcal{S}_{\delta_1}) + \sum_{i=2}^{n}(\delta_{i-1} - \delta_i)\mathcal{L}_{dec}(N^d, \boldsymbol{x}, \mathcal{S}_{\delta_i})$$

### 3.5 CIVET Algorithm

Algorithm 1 shows CIVET's training algorithm based on the $\mathcal{L}_{CIVET}$ from the Section 3.4. We specify IBP as the deterministic bounding method; however, CIVET is general for any differentiable deterministic bounding method. For each Epoch, we iterate over inputs $\boldsymbol{x}$ in dataset $\mathcal{X}$. We first compute the upper and lower bounds on the latent space distribution parameters (line 3). We then compute the weighted loss $\mathcal{L}_{CIVET}$ by first computing a support set $\mathcal{S}_{\delta_i}$ for each delta, $\delta_i$ with the FINDSUPPORT (FS) algorithm (line 4, 7). We can now use an existing deterministic bounding algorithm to compute the worst-case loss over $\theta_d$ on each $\mathcal{S}_{\delta_i}$ (line 5,8). Algorithm 2 shows a binary search algorithm for finding support sets. Note that if we reach the maximum depth we return the upper bound as it is a sound overapproximation. CIVET is the first algorithm specialized for certified training of VAEs which does not impose lipschitz restrictions on the encoder and decoder networks.

| **Algorithm 1** CIVET Algorithm | **Algorithm 2** FS($\mu_{lb}, \mu_{ub}, \sigma_{ub}, \delta, l, u, j$) |
|---|---|
| 1: **for** x $\subset \mathcal{X}$ **do** | 1: $m = (l + u)/2$ |
| 2: $\quad \boldsymbol{\mu_{lb}}, \boldsymbol{\mu_{ub}}, \boldsymbol{\sigma_{lb}}, \boldsymbol{\sigma_{ub}} \leftarrow \text{IBP}(\theta_e, \phi_t(\boldsymbol{x}))$ | 2: $s = \Phi^{-1}(m) + \Phi^{-1}(m - (1 - \delta)^{1/d_1})$ |
| 3: $\quad \mathcal{S}_{\delta_1} \leftarrow \text{FS}(\boldsymbol{\mu_{lb}}, \boldsymbol{\mu_{ub}}, \boldsymbol{\sigma_{ub}}, \delta_1, 1 - \delta_1, 1, 0)$ | 3: **if** $j = j_{max}|s = (\mu_{lb} - \mu_{ub})/\sigma_{ub}$ **then** |
| 4: $\quad \mathcal{L}_{CIVET} \leftarrow (1 - \delta_1)\mathcal{L}_{dec}(\theta_d, \boldsymbol{x}, \mathcal{S}_{\delta_1})$ | 4: $\quad$ **return** $[\mu_{lb} - \sigma_{ub}\Phi^{-1}(u),$ |
| 5: $\quad$ **for** $i \in [2, \ldots, n]$ **do** | 5: $\quad\quad \mu_{ub} + \sigma_{ub}\Phi^{-1}(u)]$ |
| 6: $\quad\quad \mathcal{S}_{\delta_i} \leftarrow \text{FS}(\boldsymbol{\mu_{lb}}, \boldsymbol{\mu_{ub}}, \boldsymbol{\sigma_{ub}}, \delta_i, 1 - \delta_i, 1, 0)$ | 6: **if** $s < (\mu_{lb} - \mu_{ub})/\sigma_{ub}$ **then** |
| 7: $\quad\quad \mathcal{L}_{CIVET} \leftarrow \mathcal{L}_{CIVET}$ | 7: $\quad$ **return** FS($\mu_{lb}, \mu_{ub}, \sigma_{ub}, \delta, m, u, j + 1$) |
| 8: $\quad\quad\quad + (\delta_{i-1} - \delta_i)\mathcal{L}_{dec}(\theta_d, \boldsymbol{x}, \mathcal{S}_{\delta_i})$ | 8: **else** |
| 9: $\quad$ Update $\theta_e, \theta_d$ using $\mathcal{L}_{CIVET}$ | 9: $\quad$ **return** FS($\mu_{lb}, \mu_{ub}, \sigma_{ub}, \delta, l, m, j + 1$) |

## 4 Evaluation

We compare CIVET to adversarial training and existing certifiably robust VAE training methods.

**Experimental Setup**. All experiments were performed on a Nvidia A100. We use the functional Lagrangian inspired probabilistic verifier proposed in Berrada et al. (2021) to perform certification. We additionally compare CIVET to baselines on empirical robustness obtained with adversarial attack methods: RAFA Liu et al. (2023) for wireless and Latent Space Attack (LSA) Kos et al. (2018)/Maximum Damage Attack (MDA) Camuto et al. (2021) for vision (see Section 4.4). We use IBP Mirman et al. (2018) for our deterministic bounding algorithm for both verification and training. We perform our experiments in two target application areas: vision and wireless. Unless otherwise specified we train CIVET with $\mathcal{D} = [0.35, 0.2, 0.05]$ as our set of $\delta s$. In Section 4.3, we experiment with different sets of $\delta s$. Results are averaged over the entire test set for each dataset and computed with $\delta = 0.05$. Certification/Attack radius is set to training $\epsilon$. Additional training parameters can be found in Appendix B.

**Wireless**. In order to achieve MIMO capabilities in 5G, base stations need to know the downlink wireless channel from their antennas to every client device. In FDD (Frequency Domain Duplex-

Table 1: Comparison of different training methods: standard training, adversarial training (PGD) and CIVET on FIRE, results reported in SNR.

| Dataset | $\epsilon$ | Training Method | Baseline | Certified | RAFA |
|---|---|---|---|---|---|
| FIRE | 15% | Standard | 17.79 dB | 4.12 dB | 5.35 dB |
| | | PGD | **17.81 dB** | 6.89 dB | 7.18 dB |
| | | CIVET | 16.58 dB | **15.02 dB** | **16.27 dB** |
| | 20% | Standard | **17.79 dB** | 1.28 dB | 4.32 dB |
| | | PGD | 17.40 dB | 4.69 dB | 5.24 dB |
| | | CIVET | 16.34 dB | **14.61 dB** | **15.98 dB** |
| | 25% | Standard | **17.79 dB** | -2.35 dB | 0.16 dB |
| | | PGD | 17.40 dB | 3.17 dB | 4.09 dB |
| | | CIVET | 15.82 dB | **12.88 dB** | **13.43 dB** |

Table 2: Comparison of different training methods: standard training, adversarial training (PGD) and CIVET on MNIST and CIFAR-10, results reported as MSE.

| Dataset | $\epsilon$ | Training Method | Baseline | Certified | LSA | MDA |
|---|---|---|---|---|---|---|
| MNIST | 0.1 | Standard | **0.0023** | 0.1426 | 0.0652 | 0.0873 |
| | | PGD | 0.0025 | 0.0648 | 0.0093 | 0.0102 |
| | | CIVET | 0.0027 | **0.0089** | **0.0065** | **0.0078** |
| | 0.3 | Standard | **0.0023** | 0.1884 | 0.0764 | 0.0922 |
| | | PGD | 0.0027 | 0.0972 | **0.0154** | 0.0386 |
| | | CIVET | 0.0031 | **0.0274** | 0.0163 | **0.0261** |
| CIFAR-10 | $\frac{2}{255}$ | Standard | **0.0041** | 0.0340 | 0.0216 | 0.0188 |
| | | PGD | **0.0041** | 0.0167 | 0.0068 | **0.0049** |
| | | CIVET | 0.0049 | **0.0055** | **0.0053** | 0.0054 |
| | $\frac{8}{255}$ | Standard | **0.0041** | 0.2098 | 0.0562 | 0.0801 |
| | | PGD | 0.0043 | 0.0760 | 0.0173 | **0.0093** |
| | | CIVET | 0.0062 | **0.0153** | **0.0087** | 0.0124 |

ing) systems, dominant in the United States, the client devices measure the wireless channel using extra preamble symbols transmitted by the base station and send it as feedback to the base station. However, this feedback is unsustainable and causes huge spectrum waste. Recent work Liu et al. (2021) proposed FIRE which uses an end-to-end ML based approach to predict the downlink channels. For this paper, we choose to evaluate against FIRE because it shows SOTA performance, uses a VAE architecture, and Liu et al. (2023) shows that FIRE is vulnerable to real-world adversarial attacks. Errors in downlink channel estimates reduce the communication efficiency of multi-antenna systems (e.g., MIMO). Robustly training FIRE will allow it to be safely deployed in real-world systems. Additional details on FIRE and the choice of VAEs can be found in Appendix E.

For our wireless experiments, we do a best-effort re-implementation of FIRE Liu et al. (2021). We borrow the data and neural networks used by Liu et al. (2023). The VAE has 7 linear layers in both the encoder and decoder networks with a 50 dimensional latent space. Liu et al. (2023) collected 10,000 data points by moving the antenna randomly in a 10m by 7m space, and is composed of many reflectors (like metal cupboards, white-boards, etc.) and obstacles. We use the same 8:2 train/test split. We also adopt the same adversarial budget used by Liu et al. (2023): the perturbation is allowed a percentage of the average amplitude of the benign channel estimates. We use Signal-Noise Ratio (SNR) to report performance for wireless, similar to (Liu et al. (2021; 2023)).

**Vision**. We consider two popular image recognition datasets: MNIST Deng (2012) and CIFAR10 Krizhevsky et al. (2009). We use a variety of challenging $l_\infty$ perturbation bounds common in verification/robust training literature Xu et al. (2021); Wang et al. (2021); Singh et al. (2019; 2018a); Shi et al. (2021); Mueller et al. (2022); Mao et al. (2023). We use a VAE with 3 convolutional and 1 linear layer for both the encoder and decoder. For MNIST we use a 32 dimensional latent space and for CIFAR-10 we use a 64 dimensional latent space. Both MNIST and CIFAR10 have a test set of 10,000 images. We compare the performance for both datasets using Mean Squared Error (MSE).

Table 3: Comparison of CIVET and Lipschitz VAEs Barrett et al. (2022) on MNIST and CIFAR-10, MSE is reported.

| Dataset | Architecture | $\epsilon$ | Training Method | Baseline | Certified | LSA | MDA |
|---------|--------------|-----------|-----------------|----------|-----------|-----|-----|
| MNIST | FC | 0.1 | Lipschitz | 0.0049 | 0.0253 | 0.0168 | 0.0211 |
| | | | CIVET | **0.0038** | **0.0230** | **0.0114** | **0.0197** |
| | | 0.3 | Lipschitz | 0.0064 | **0.0486** | **0.0366** | 0.0409 |
| | | | CIVET | **0.0043** | 0.0507 | 0.0412 | **0.0359** |
| CIFAR-10 | Conv | $\frac{2}{255}$ | Lipschitz | 0.0083 | 0.0105 | 0.0089 | 0.0096 |
| | | | CIVET | **0.0049** | **0.0055** | **0.0053** | **0.0054** |
| | | $\frac{8}{255}$ | Lipschitz | 0.0112 | 0.0267 | 0.0178 | 0.0252 |
| | | | CIVET | **0.0062** | **0.0153** | **0.0087** | **0.0124** |

## 4.1 MAIN RESULTS

We compare CIVET to standard training and adversarial training on FIRE in Table 1 and vision in Table 2. Across all datasets and $\epsilon$s we observe that CIVET obtains significantly better certified performance (e.g. 13.88 dB vs -2.35 dB for FIRE with $\epsilon = 25\%$ and 0.0089 vs 0.01426 for MNIST with $\epsilon = 0.1$). CIVET obtains comparable performance on baseline metrics. Notably, CIVET still outperforms traditional non-ML baselines Liu et al. (2021). Our results indicate that CIVET obtains significantly better certified performance while losing some baseline performance, inline with certified training approaches for other NN architectures Shi et al. (2021); Gowal et al. (2018).

## 4.2 COMPARISON TO LIPSCHITZ VAES

In this section, we compare CIVET to Lipschitz VAEs proposed by Barrett et al. (2022). A network $f : \mathbb{R}^{d_{in}} \rightarrow \mathbb{R}^{d_{out}}$ is Lipschitz continuous if for all $\mathbf{x}_1, \mathbf{x}_2 \in \mathbb{R}^{d_{in}}, ||f(\mathbf{x}_1) - f(\mathbf{x}_2)|| \leq M||\mathbf{x}_1 - \mathbf{x}_2||$ for constant $M \in \mathbb{R}^+$. The least $M$ for which this holds is the Lipschitz constant of $f$. Barrett et al. (2022) trains Lipschitz-constrained VAEs with fixed variances in the latent space. Barrett et al. (2022) only reports results for fully connected MNIST VAEs. For MNIST, we follow their network architecture with a latent space of 10 and 3 fully connected layers. For CIFAR-10, we use our convolutional network architecture with a latent space of 64. Barrett et al. (2022) only trains networks with a GroupSort activation. Anil et al. (2019) shows that Lipschitz constrained networks are limited in expressivity when using non-gradient norm preserving activations such as sigmoid or ReLU introducing GroupSort as an alternative (it can be shown that GroupSort is equivalent to ReLU when the group has size 2 Anil et al. (2019)). Following Barrett et al. (2022) we compare their networks with GroupSort to our networks with ReLU. Table 3 provides detailed results on the comparison of Lipschitz VAEs and CIVET. In some cases (CIFAR-10) CIVET almost doubles the performance of Lipschitz VAEs (0.0062 vs 0.0112 8/255 baseline, 0.0055 vs 0.0105 2/255 certified). For MNIST with $\epsilon - 0.3$ Lipschitz VAE outperforms CIVET for certified performance and LSA performance; however, CIVET still greatly outperforms in baseline performance (0.0043 vs 0.0064). Therefore, CIVET is better on almost all benchmarks while generalizing to more network architectures.

## 4.3 ABLATION STUDIES

For our experiments we choose $\mathcal{D} = [0.35, 0.2, 0.05]$ or $|\mathcal{D}| = 3$ and $\eta = 0.15$. In this ablation study, we compare results on FIRE at 25% perturbation budget while varying $\mathcal{D}$. In Appendix H, Figure 3 shows the average standard and certified SNR for networks trained using CIVET with varying $\mathcal{D}$s. In both graphs we see increasing standard SNR as either the size of $\mathcal{D}$ increases or the difference between each of the $\delta$s increases, but observe that certified SNR decreases past a certain point. Note that on the left side when $|\mathcal{D}| = 5$ the largest $\delta = 0.65$ and on the right side when $\eta = 0.25$ the largest $\delta = 0.55$. We hypothesize that adding such a large $\delta$ decreases regularization leading to improved accuracy, but no longer captures a significant portion of the distribution leading to decreased certified SNR. We leave further study of $\mathcal{D}$ selection to future work. We also test the affect of combining CIVET with SABR Mueller et al. (2022) in Appendix G.

### 4.4 PERFORMANCE AGAINST ATTACKS

Liu et al. (2023) proposes RAFA (RAdio Frequency Attack), the first hardware-implemented adversarial attack against ML-based wireless systems. Specifically, Liu et al. (2023) shows that RAFA can severely degrade the performance of FIRE in the real-world. For our vision datasets we compare our results against SOTA VAE attacks. LSA Kos et al. (2018) tries to maximize the KL divergence in the latent space, while MDA Camuto et al. (2021) maximizes the reconstruction distance. We compare CIVET against the baselines on these attack methods in Tables 1,2,3. CIVET outperforms baselines on most benchmarks, when CIVET underperforms it is tied to its decrease in baseline accuracy.

### 4.5 RUNTIME ANALYSIS

For CIFAR10 and $\epsilon = 8/255$, standard training took 45 minutes, adversarial training took 68 minutes, Lipschitz VAE took 73 minutes, and CIVET training took 296 minutes (runtimes for FIRE and MNIST can be found in Appendix B). The main difference comes from taking 3 passes of the decoder network. CIVET using $\mathcal{D} = [0.05]$ on CIFAR-10 takes 86 minutes significantly closer to baseline timings while obtaining a baseline performance of 0.0075 and certified performance of 0.0219 still outperforming Lipschitz VAE on both metrics and all baselines in certified performance.

## 5 RELATED WORK

**Deterministic DNN Verification**. Neural network verification is generally NP-complete Katz et al. (2017a) so most existing methods trade precision for scalability. Single-input robustness can be deterministically analyzed via abstract interpretation Singh et al. (2019); Banerjee et al. (2024c) or via optimization using linear programming (LP) De Palma et al. (2021); Müller et al. (2022), mixed integer linear programming (MILP) Singh et al. (2018b); Tjeng et al. (2018). Recent studies have extended verification to incremental settings Ugare et al. (2023; 2024), hyperproperties Banerjee & Singh (2024); Suresh et al. (2024); Banerjee et al. (2024b), and DNN interpretation Banerjee et al. (2024a).

**Probablistic DNN Verification**. Cardelli et al. (2019); Michelmore et al. (2019) give statistical confidence bounds on the robustness of Bayesian Neural Networks (BNNs). Wicker et al. (2020) gives certified guarantees on the probabilistic safety of BNNs. Berrada et al. (2021) introduces the functional Lagrangian to give a general framework for giving certified guarantees on probabilistic specifications, their formulation is general and can handle stochastic inputs, BNNs, and VAEs.

**Certified Training of Deterministic DNNs**. Shi et al. (2021); Mirman et al. (2018); Balunović & Vechev (2020); Zhang et al. (2019) are well-known approaches for certified training of standard DNNs. More recent works Mueller et al. (2022); Xiao et al. (2019); Fan & Li (2021); Palma et al. (2024) integrate adversarial and certified training techniques to achieve state-of-the-art performance in both robustness and clean accuracy.

**Certified Training for Stochastic DNNs**. Wicker et al. (2021) proposes an IBP-based certified training method for BNNs. Wicker et al. (2021) samples the parameters (weights and biases) of BNNs to generate a set of deterministic networks. They then apply IBP to each of these deterministic networks during training. While CIVET utilizes the same high-level idea of reducing certified training for stochastic networks to that of deterministic networks, it exploits the unique structure of VAEs. CIVET is also general for arbitrary differentiable deterministic bounding methods. Barrett et al. (2022) trains certifiably robust VAEs by imposing lipschitz conditions and a fixed variance. CIVET removes these restrictions handling a more general class of VAEs.

## 6 LIMITATIONS/CONCLUSION

We discuss the limitations of CIVET in Appendix F. In this paper, we introduce a novel certified training method for VAEs called CIVET. CIVET is based on our theoretical analysis of the VAE robustness problem and based on the key insight that it is possible to find support sets which over-approximate this loss, and that these support sets can then be analyzed using existing deterministic neural network bounding algorithms. CIVET lays the groundwork for a new set of certified training methods for VAEs and other stochastic neural network architectures.

## REPRODUCIBILITY STATEMENT

To assist with reproducibility and further research we have released the code used for our results publicly. Section 4 gives details on our evaluation which is supplemented by Appendix B. Appendix A contains full proofs of all Theorems and Lemmas stated in the paper, and all assumptions made have been stated in the main body of the paper. Section F also gives an overview of some additional assumptions made.

## ACKNOWLEDGMENT

We thank the anonymous reviewers for their insightful comments. This work was supported in part by NSF Grants No. CCF-2238079, CCF-2316233, CNS-2148583.

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

# A  ADDITIONAL PROOFS

**Theorem 1.** *For a VAE with encoder $N^e$, decoder $N^d$, local input region $\phi_t(\boldsymbol{x_0})$, error function $M$ and probability threshold $(1 - \delta)$, if $\mathcal{Z} = \{Z \mid Z = N^e(\boldsymbol{x}), \boldsymbol{x} \in \phi_t(\boldsymbol{x_0})\}$ then for any support set $\mathcal{S}$ for $\mathcal{Z}$ the worst case error $\mathcal{L}_w(N^e, N^d, \boldsymbol{x_0}) \lesssim T_{ub}(\mathcal{S})$ where $T_{ub}(\mathcal{S}) = \max_{z \in \mathcal{S}} M(N^d(\boldsymbol{z}))$.*

*Proof.* Let, $I_{t_0}(\boldsymbol{y}) = (M(\boldsymbol{y}) \leq t_0)$ be an indicator where $t_0 = T_{ub}(\mathcal{S})$. Then for any output distribution $Y = N^d(Z)$, $P(M(Y) \leq t_0)$ can be written as $\int_{\mathbb{R}^{d_l}} I_{t_0}(N^d(\boldsymbol{z})) \times f_Z(\boldsymbol{z}) d\boldsymbol{z}$ where $Z \in \mathcal{Z}$ and $f_Z$ is the probability density function of $Z$. Note here $\boldsymbol{z} \in \mathbb{R}^{d_l}$ are vectors and $d\boldsymbol{z} = dz_1 \ldots dz_{d_l}$. Now since the support set $\mathcal{S} \subseteq \mathbb{R}^{d_l}$ and $I_{t_0}(N^d(\boldsymbol{z})) \times f_Z(\boldsymbol{z}) \geq 0$ for all $\boldsymbol{z} \in \mathbb{R}^{d_l}$

$$
\begin{aligned}
P(M(Y) \leq t_0) &= \int_{\mathbb{R}^{d_l}} I_{t_0}(N^d(\boldsymbol{z})) \times f_Z(\boldsymbol{z}) d\boldsymbol{z} \\
&\geq \int_{\mathcal{S}} I_{t_0}(N^d(\boldsymbol{z})) \times f_Z(\boldsymbol{z}) d\boldsymbol{z} \quad \text{given } \mathcal{S} \subseteq \mathbb{R}^{d_l} \text{ and } I_{t_0}(N^d(\boldsymbol{z})) \times f_Z(\boldsymbol{z}) \geq 0 \\
&\geq \int_{\mathcal{S}} f_Z(\boldsymbol{z}) d\boldsymbol{z} \quad t_0 = T_{ub}(\mathcal{S}) \text{ implies } I_{t_0}(N^d(\boldsymbol{z})) = 1 \forall \boldsymbol{z} \in \mathcal{S} \\
&\geq (1 - \delta) \quad \mathcal{S} \text{ is support so } P(Z \in \mathcal{S}) \geq (1 - \delta)
\end{aligned}
\tag{3}
$$

From Eq. 1 and 3, for any output distribution $Y = N^d(Z)$, $T(Y) \leq t_0$. Now, given for all possible output distributions $Y \in \mathcal{Y}$ as $T(Y) \leq t_0$, the worst-case loss $\mathcal{L}_w(N^e, N^d, \boldsymbol{x_0}) = \max_{Y \in \mathcal{Y}} T(Y) \leq t_0 = T_{ub}(\mathcal{S})$. Note in all cases we assume the indicator function $I_{t_0}(\boldsymbol{y})$ is well behaved and both the integrals $\int_{\mathcal{S}} I_{t_0}(N^d(\boldsymbol{z})) \times f_Z(\boldsymbol{z}) d\boldsymbol{z}$ and $\int_{\mathbb{R}^{d_l}} I_{t_0}(N^d(\boldsymbol{z})) \times f_Z(\boldsymbol{z}) d\boldsymbol{z}$ is well defined. $\qquad \square$

Let $\mathcal{C}(\mu, \sigma, u, l) = \Phi_{\mu, \sigma}(u) - \Phi_{\mu, \sigma}(l)$ where $\Phi_{\mu, \sigma} : \mathbb{R} \to [0, 1]$ is the cdf of the following gaussian distribution $\mathcal{N}(\mu, \sigma)$.

**Lemma 2.** *For a fixed mean $\mu \in [\mu_{lb}, \mu_{ub}]$ and bounds $l \leq \mu_{lb}$, $\mu_{ub} \leq u$, $\sigma < \sigma_{ub} \implies \mathcal{C}(\mu, \sigma, u, l) > \mathcal{C}(\mu, \sigma_{ub}, u, l)$.*

*Proof.*

$$
\begin{aligned}
\mathcal{C}(\mu, \sigma, u, l) &= \Phi_{\mu, \sigma}(u) - \Phi_{\mu, \sigma}(l) \\
&= \Phi_{0,1}\left(\frac{u - \mu}{\sigma}\right) - \Phi_{0,1}\left(\frac{l - \mu}{\sigma}\right) \\
&= \Phi_{0,1}\left(\frac{u - \mu}{\sigma}\right) + \Phi_{0,1}\left(\frac{\mu - l}{\sigma}\right) - 1 \quad \text{using } \Phi_{0,1}(x) = 1 - \Phi_{0,1}(-x)
\end{aligned}
\tag{4}
$$

Now, since for any $\mu \in [\mu_{lb}, \mu_{ub}]$ and $l \leq \mu_{lb}$, $(\mu - l) \geq 0$, $\sigma < \sigma_{ub} \implies \frac{\mu - l}{\sigma_{ub}} < \frac{\mu - l}{\sigma}$. $\Phi$ is monotonically increasing. Hence, $\Phi_{0,1}\left(\frac{\mu - l}{\sigma_{ub}}\right) < \Phi_{0,1}\left(\frac{\mu - l}{\sigma}\right)$. Similarly, we show that $\Phi_{0,1}\left(\frac{u - \mu}{\sigma_{ub}}\right) < \Phi_{0,1}\left(\frac{u - \mu}{\sigma}\right)$. This gives us

$$
\Phi_{0,1}\left(\frac{\mu - l}{\sigma_{ub}}\right) + \Phi_{0,1}\left(\frac{u - \mu}{\sigma_{ub}}\right) - 1 < \Phi_{0,1}\left(\frac{\mu - l}{\sigma}\right) + \Phi_{0,1}\left(\frac{u - \mu}{\sigma}\right) - 1
$$
$$
\mathcal{C}(\mu, \sigma_{ub}, u, l) < \mathcal{C}(\mu, \sigma, u, l) \quad \text{Using Eq. 4}
$$

$\qquad \square$

**Lemma 3.** *For a fixed $\sigma \in [\sigma_{lb}, \sigma_{ub}]$ and bounds $l = \mu_{lb} - \zeta$, $u = \mu_{ub} + \zeta$ with $\zeta \geq 0$, for any $\mu \in [\mu_{lb}, \mu_{ub}]$*

$$
\mathcal{C}(\mu, \sigma, u, l) \geq \mathcal{C}(\mu_{lb}, \sigma, u, l) = \mathcal{C}(\mu_{ub}, \sigma, u, l)
$$

*Proof.* First, we show $\mathcal{C}(\mu, \sigma, u, l)$ is symmetric around $\mu_0 = \frac{\mu_{lb} + \mu_{ub}}{2}$ i.e. $\mathcal{C}(\mu_0 + d, \sigma, u, l) = \mathcal{C}(\mu_0 - d, \sigma, u, l)$ for all $d \in [0, w/2]$ where $w = (\mu_{ub} - \mu_{lb})$ is the width of the interval $[\mu_{lb}, \mu_{ub}]$.

$$\mathcal{C}(\mu_0 + d, \sigma, u, l) = \Phi_{\mu_0+d,\sigma}(u) - \Phi_{\mu_0+d,\sigma}(l)$$

$$= \Phi_{0,1}\left(\frac{u - \mu_0 - d}{\sigma}\right) - \Phi_{0,1}\left(\frac{l - \mu_0 - d}{\sigma}\right) \tag{5}$$

Now, $u - \mu_0 - d = \mu_{ub} + \zeta - \frac{\mu_{ub} + \mu_{lb}}{2} - d = \frac{\mu_{ub} + \mu_{lb}}{2} - \mu_{lb} + \zeta - d = -(l - \mu_0 + d)$.
$l - \mu_0 - d = \mu_{lb} - \zeta - \frac{\mu_{ub} + \mu_{lb}}{2} - d = \frac{\mu_{ub} + \mu_{lb}}{2} - \mu_{ub} - \zeta - d = -(u - \mu_0 + d)$

Using Eq. 5

$$\mathcal{C}(\mu_0 + d, \sigma, u, l) = \Phi_{0,1}\left(\frac{u - \mu_0 - d}{\sigma}\right) - \Phi_{0,1}\left(\frac{l - \mu_0 - d}{\sigma}\right)$$

$$= \Phi_{0,1}\left(-\frac{l - \mu_0 + d}{\sigma}\right) - \Phi_{0,1}\left(-\frac{u - \mu_0 + d}{\sigma}\right)$$

$$= \Phi_{0,1}\left(\frac{u - \mu_0 + d}{\sigma}\right) - \Phi_{0,1}\left(\frac{l - \mu_0 + d}{\sigma}\right) \quad \text{using } \Phi_{0,1}(x) = 1 - \Phi_{0,1}(-x)$$

$$= \mathcal{C}(\mu_0 - d, \sigma, u, l) \tag{6}$$

Eq. 6 proves $\mathcal{C}(\mu_{lb}, \sigma, u, l) = \mathcal{C}(\mu_{ub}, \sigma, u, l)$ for $d = \frac{\mu_{ub} - \mu_{lb}}{2}$. Now we show that for $d_1, d_2 \in [0, \frac{\mu_{ub} - \mu_{lb}}{2}]$ if $d_1 \leq d_2$ then $\mathcal{C}(\mu_0 + d_1, \sigma, u, l) \geq \mathcal{C}(\mu_0 + d_2, \sigma, u, l)$.

$$\Phi_{0,1}\left(\frac{u - \mu_0 - d_1}{\sigma}\right) - \Phi_{0,1}\left(\frac{u - \mu_0 - d_2}{\sigma}\right) = \Phi_{0,1}\left(\frac{d_2 - (u - \mu_0)}{\sigma}\right) - \Phi_{0,1}\left(\frac{d_1 - (u - \mu_0)}{\sigma}\right)$$

$$= \frac{1}{\sqrt{2\pi}\sigma} \int_{d_1}^{d_2} \exp\left(-\frac{(x - (u - \mu_0))^2}{2\sigma^2}\right) dx \tag{7}$$

Similarly

$$\Phi_{0,1}\left(\frac{l - \mu_0 - d_1}{\sigma}\right) - \Phi_{0,1}\left(\frac{l - \mu_0 - d_2}{\sigma}\right) = \frac{1}{\sqrt{2\pi}\sigma} \int_{d_1}^{d_2} \exp\left(-\frac{(x - (l - \mu_0))^2}{2\sigma^2}\right) dx \tag{8}$$

Eq 7 - Eq 8 gives us

$$\mathcal{C}(\mu_0 + d_1, \sigma, u, l) - \mathcal{C}(\mu_0 + d_2, \sigma, u, l) = \frac{1}{\sqrt{2\pi}\sigma} \int_{d_1}^{d_2} f(x) dx \tag{9}$$

$$\text{where } f(x) = \left(\exp\left(-\frac{(x - (u - \mu_0))^2}{2\sigma^2}\right) - \exp\left(-\frac{(x - (l - \mu_0))^2}{2\sigma^2}\right)\right)$$

Next, we show that $f(x) \geq 0$ for all $x \in [d_1, d_2]$

$$(x - (u - \mu_0))^2 = (w + \zeta - x)^2 \leq (w + \zeta + x)^2 = (x - (l - \mu_0))^2 \quad \text{since } 0 \leq d_1 \leq x \leq d_2 \leq w + \zeta$$

$$\implies \left(\exp\left(-\frac{(x - (u - \mu_0))^2}{2\sigma^2}\right) \geq \exp\left(-\frac{(x - (l - \mu_0))^2}{2\sigma^2}\right)\right)$$

$$\implies f(x) \geq 0 \quad \text{where } x \in [d_1, d_2]$$

$$\implies \mathcal{C}(\mu_0 + d_1, \sigma, u, l) - \mathcal{C}(\mu_0 + d_2, \sigma, u, l) = \int_{d_1}^{d_2} f(x) \geq 0 \quad \text{from Eq. 9}$$

$$\implies \mathcal{C}(\mu_0 + d_1, \sigma, u, l) \geq \mathcal{C}(\mu_0 + d_2, \sigma, u, l)$$

This completes the proof because for any $d \in [0, \frac{\mu_{ub} - \mu_{lb}}{2}]$, $\mathcal{C}(\mu_0 + d, \sigma, u, l) \geq \mathcal{C}(\mu_{ub}, \sigma, u, l)$ and subsequently $\mathcal{C}(\mu_0 - d, \sigma, u, l) \geq \mathcal{C}(\mu_{ub}, \sigma, u, l)$. $\square$

**Lemma 1.** *Given bounds* $\mu_{lb}, \mu_{ub}, \sigma_{lb}, \sigma_{ub} \in \mathbb{R}$, *probability threshold* $(1 - \delta)$, *and* $\zeta \in \mathbb{R}^+$. *Let* $\mathcal{C}(\mu, \sigma, u, l) = \Phi_{\mu,\sigma}(u) - \Phi_{\mu,\sigma}(l)$. *Then* $\forall \mu \in [\mu_{lb}, \mu_{ub}], \sigma \in [\sigma_{lb}, \sigma_{ub}]$,

$$\mathcal{C}(\mu, \sigma, \mu_{ub} + \zeta, \mu_{lb} - \zeta) \geq \mathcal{C}(\mu_{ub}, \sigma_{ub}, \mu_{ub} + \zeta, \mu_{lb} - \zeta)$$

*Furthermore,*

$$\mathcal{C}(\mu_{lb}, \sigma_{ub}, \mu_{ub} + \zeta, \mu_{lb} - \zeta) = \mathcal{C}(\mu_{ub}, \sigma_{ub}, \mu_{ub} + \zeta, \mu_{lb} - \zeta)$$

*Proof.*

$$\mathcal{C}(\mu, \sigma, \mu_{ub} + \zeta, \mu_{lb} - \zeta) \geq \mathcal{C}(\mu, \sigma_{ub}, \mu_{ub} + \zeta, \mu_{lb} - \zeta) \text{ Using lemma 2}$$
$$\geq \mathcal{C}(\mu_{ub}, \sigma_{ub}, \mu_{ub} + \zeta, \mu_{lb} - \zeta) \text{ Using lemma 3}$$

The proof of $\mathcal{C}(\mu_{lb}, \sigma_{ub}, \mu_{ub} + \zeta, \mu_{lb} - \zeta) = \mathcal{C}(\mu_{ub}, \sigma_{ub}, \mu_{ub} + \zeta, \mu_{lb} - \zeta)$ comes from Eq. 6.     □

**Theorem 2.** *Given $\mathcal{Z} = \{\mathcal{N}(\mu, \sigma) \mid \mu \in [\mu_{lb}, \mu_{ub}], \sigma \in [\sigma_{lb}, \sigma_{ub}]\}$, probability threshold $(1 - \delta)$, then $[l, u] = [\mu_{lb} - \sigma_{ub}\Phi^{-1}(p_0), \mu_{ub} + \sigma_{ub}\Phi^{-1}(p_0)]$ satisfies Equation 2. Where*

$$p_0 = \min_{p \in [(1-\delta), 1]} \left[ \Phi^{-1}(p) + \Phi^{-1}(p - (1 - \delta)) \geq \frac{\mu_{lb} - \mu_{ub}}{\sigma_{ub}} \right]$$

.

*Proof.* We would like to show that $\forall \mu \in [\mu_{lb}, \mu_{ub}], \sigma \in [\sigma_{lb}, \sigma_{ub}].\mathcal{C}(\mu, \sigma, \mu_{ub} + \sigma_{ub}\Phi^{-1}(p_0), \mu_{lb} - \sigma_{ub}\Phi^{-1}(p_0)) \geq (1 - \delta)$. By Lemma 1 we have $\forall \mu \in [\mu_{lb}, \mu_{ub}], \sigma \in [\sigma_{lb}, \sigma_{ub}]$

$$\mathcal{C}(\mu, \sigma, \mu_{ub} + \sigma_{ub}\Phi^{-1}(p_0), \mu_{lb} - \sigma_{ub}\Phi^{-1}(p_0)) \geq \mathcal{C}(\mu_{ub}, \sigma_{ub}, \mu_{ub} + \sigma_{ub}\Phi^{-1}(p_0), \mu_{lb} - \sigma_{ub}\Phi^{-1}(p_0))$$

Therefore, it is sufficient to prove that

$$\mathcal{C}(\mu_{ub}, \sigma_{ub}, \mu_{ub} + \sigma_{ub}\Phi^{-1}(p_0), \mu_{lb} - \sigma_{ub}\Phi^{-1}(p_0)) \geq (1 - \delta)$$

We can start by expanding $\mathcal{C}(\mu_{ub}, \sigma_{ub}, \mu_{ub} + \sigma_{ub}\Phi^{-1}(p), \mu_{lb} - \sigma_{ub}\Phi^{-1}(p))$

$$
\begin{aligned}
\mathcal{C}(\mu_{ub}, \sigma_{ub}, & \mu_{ub} + \sigma_{ub}\Phi^{-1}(p), \mu_{lb} - \sigma_{ub}\Phi^{-1}(p)) \\
&= \Phi_{\mu_{ub}, \sigma_{ub}}(\mu_{ub} + \sigma_{ub}\Phi^{-1}(p)) - \Phi_{\mu_{ub}, \sigma_{ub}}(\mu_{lb} - \sigma_{ub}\Phi^{-1}(p)) \\
&= \Phi\left(\frac{\sigma_{ub}\Phi^{-1}(p))}{\sigma_{ub}}\right) - \Phi\left(\frac{\mu_{lb} - \sigma_{ub}\Phi^{-1}(p) - \mu_{ub}}{\sigma_{ub}}\right) \\
&= \Phi\left(\Phi^{-1}(p)\right) - \Phi\left(\frac{\mu_{lb} - \mu_{ub}}{\sigma_{ub}} - \Phi^{-1}(p)\right) \\
&= p - \Phi\left(\frac{\mu_{lb} - \mu_{ub}}{\sigma_{ub}} - \Phi^{-1}(p)\right)
\end{aligned}
$$

Now, we want $p$ s.t.

$$
\begin{aligned}
p - \Phi\left(\frac{\mu_{lb} - \mu_{ub}}{\sigma_{ub}} - \Phi^{-1}(p)\right) &\geq (1 - \delta) \\
p - (1 - \delta) &\geq \Phi\left(\frac{\mu_{lb} - \mu_{ub}}{\sigma_{ub}} - \Phi^{-1}(p)\right) \\
\Phi^{-1}(p - (1 - \delta)) &\geq \frac{\mu_{lb} - \mu_{ub}}{\sigma_{ub}} - \Phi^{-1}(p) \\
\Phi^{-1}(p - (1 - \delta)) + \Phi^{-1}(p) &\geq \frac{\mu_{lb} - \mu_{ub}}{\sigma_{ub}}
\end{aligned}
$$

Note that the $\Phi^{-1}$ is a strictly increasing function. Since $p_0$ is defined as the min $p$ that satisfies this condition, we have proved the Theorem.     □

# B EVALUATION DETAILS

We implemented CIVET in PyTorch Paszke et al. (2019). For additional details see our codebase. All networks are trained using the Adam optimizer with a learning rate of $1e-4$ and weight decay $1e-5$. All networks are trained with 100 epochs. We use a batch size of 16 for MNIST and 32 for FIRE and CIFAR-10.

## B.1 TRAINING METHODS

**Standard**. Standard training is done using a combination of KL divergence loss on the mean/standard deviation and reconstruction loss on the output.

**PGD**. PGD training is done by mixing standard loss with an adversarial loss. The adversarial loss is computed by first computing an adversarial perturbation using PGD. PGD is instantiated with the same KL divergence/reconstruction loss combination as standard training, we use a step size equal to $0.1 \cdot \epsilon$ and perform 10 iterations. This adversary is added to the input and then fed through the network to compute the loss.

**CIVET**. Sticking with standard IBP protocols, we start by warming up with standard loss for the first 250 iterations (250 batches). For the next 250 batches we linearly scale $\epsilon$ from 0 and add the CIVET loss to the standard loss. After these warmup stages we compute CIVET loss and add it to standard loss.

**CIVET-SABR**. We perform the same steps as CIVET training but first compute a maximum damage attack (MDA) on the input using a radius of $(1 - \tau) \cdot \epsilon$ setting $\tau = 0.1$. We then compute a smaller ball around this adversarial example with radius $\tau \cdot \epsilon$ and perform normal CIVET training.

## B.2 DATASETS

**Wireless**. We use the same VAE architecture as Liu et al. (2021). The VAE encoder has 7 linear layers: starting with a hidden size of 1024 going down by a factor of 2 each time, the VAE decoder as the same sizes in the opposite direction. We use a latent dimension of 50. All layers use a LeakyReLU activations with a tanh activation at the end. With $\epsilon = 25\%$, standard training took 18 minutes, adversarial training took 25 minutes, and CIVET training took 118 minutes.

**Vision**. We use convolutional layers with a kernel size of 5, stride of 2, and a padding of 1. We use three convolutional layers starting with 16 (64 for CIFAR) channels and doubling each time. For MNIST we use a latent dimension of 32 and for CIFAR10 we use a latent dimension of 64. All layers use ReLU activations with a sigmoid activation at the end. With $\epsilon = 0.3$ for MNIST, standard training took 16 minutes, adversarial training took 18 minutes, and CIVET training took 93 minutes. For the fully connected MNIST network, there are 2 hidden layers with size 512 and 1 with size 10 for both the encoder and decoder. Lipschitz VAE with GroupSort activations took 24 minutes to train and /method with ReLU activations took 41 minutes to train.

# C EXAMPLE FOR SAMPLE SELECTION

This section elaborates on the key steps of the support selection algorithm with an example. We consider the one-dimensional case for simplicity. Let the ranges of the mean and variance be the intervals $[\mu_{lb}, \mu_{ub}]$ and $[\sigma_{lb}, \sigma_{ub}]$, where $\mu_{lb} = 0.0$, $\mu_{ub} = 1.0$, $\sigma_{lb} = 1.0$, and $\sigma_{ub} = 2.0$. For a fixed $\delta = 0.05$ (i.e., a $95\%$ confidence level), the algorithm computes an interval $[l, u]$ that captures at least $95\%$ of the probability for all possible Gaussian distributions with $\mu \in [\mu_{lb}, \mu_{ub}]$ and $\sigma \in [\sigma_{lb}, \sigma_{ub}]$.

From Lemma 1, for any $l \leq \mu_{lb}$ and $\mu_{lb} \leq u$, we know that the distributions specified by $\mu_{lb}, \sigma_{ub}$ and $\mu_{ub}, \sigma_{ub}$ capture the least probability among all possible distributions. Additionally, since $(1 - \delta) > 0.5$, both $l$ and $u$ automatically satisfy the constraints $l \leq \mu_{lb}$ and $\mu_{lb} \leq u$. Therefore, finding the support set $[l, u]$ only requires ensuring that both the distributions specified by $\mu_{lb}, \sigma_{ub}$ and $\mu_{ub}, \sigma_{ub}$ capture at least $(1 - \delta)$ probability. Finally, the bounds $l$ and $u$ are obtained by applying Theorem 2, where $l = 0.0 - 3.54 = -3.54$ and $u = 1.0 + 3.54 = 4.54$.

## D    JUSTIFICATION OF THE WEIGHTING SCHEME

In this section, we explain the rationale for using the weighing scheme to define the CIVET loss (Definition. 2). For a Gaussian distribution, most of the probability mass is concentrated around the mean. This means that the support set $[l, u]$ (or its length in the 1D case) grows significantly faster than the additional probability it captures. For instance, for any Gaussian distribution with parameters $(\mu, \sigma)$, the interval $[\mu - 3\sigma, \mu + 3\sigma]$ captures 99.73% of the probability mass, while extending it to $[\mu - 4\sigma, \mu + 4\sigma]$ increases the coverage to only 99.99%. Thus, the support length must increase by 33% to capture just an additional 0.26% of the probability. Moreover, IBP with larger intervals accumulates greater approximation error. Using the suggested weighting scheme, $\sum_i (1 - \delta_i)\mathcal{L}_i$, would unnecessarily assign very high weights to larger intervals, which generally capture negligible additional probability mass compared to the smaller intervals (i.e., $[l_{i-1}, u_{i-1}]$ corresponding to $(1 - \delta_{i-1})$.

## E    ADDITIONAL DISCUSSION ON FIRE

FIRE Liu et al. (2021) uses a VAE architecture to predict downlink channels inspired by a physics-level intuition that both uplink and downlink channels are generated by the same process from the underlying physical environment. A VAE can first infer a latent low-dimensional representation of the underlying process of channel generation by observing samples of the uplink channel, and then generate the downlink channel by sampling in this low-dimensional space. This allows the VAE to embed real-world effects in the latent space and therefore capture the generative process more accurately.

For FIRE we compare performance using SNR. The SNR of the predicted channel $H$ can be computed by comparing to the ground truth channel $H_{gt}$ by using the following:

$$SNR(H, H_{gt}) = -10log_{10}\left(\frac{||H - H_{gt}||^2}{||H_{gt}||^2}\right) \qquad (10)$$

## F    LIMITATIONS

CIVET currently restricts support set selection to the interval domain; however, more precise support set selection may be possible with the zonotope domain (as commonly used in DNN verification Singh et al. (2018a)) or the octogon domain (as commonly used in probabalistic programming Sankaranarayanan et al. (2013)), we leave this for future work to explore. CIVET focuses on the gaussian distribution as commonly used in VAEs; however, it is possible to generalize our method to families of distributions as in Berrada et al. (2021). CIVET assumes an adversary is attacking with single-input adversarial examples; however, the RAFA attack used by Liu et al. (2023) to attack FIRE is a universal adversarial perturbation which is a weaker attack model. It may be possible to obtain better performance by adjusting CIVET to handle universal attacks instead. Recent work in attacks Moosavi-Dezfooli et al. (2017); Liu et al. (2023); Xu & Singh (2022), certification Banerjee et al. (2024c); Banerjee & Singh (2024); Suresh et al. (2024) and certified training for UAPs Xu & Singh (2024) suggest that improved accuracy with similar robustness is possible when defending against these weaker attack models; however, we leave this for future work.

## G    CIVET-SABR

We present results using SABR Mueller et al. (2022), a state-of-the-art certified training method. SABR works by first computing an adversarial attack and then propagating a small adversarial box around that adversarial attack rather than propogating the entire input region. Mueller et al. (2022) note that although this training method is now unsound the approximation errors are significantly less resulting in lowered regularization and increased perfornace. We compute a maximum damage attack to find an adversarial example in the input space. We then compute a smaller bounding box ($\tau = 0.1$, i.e. $L_\infty$ ball with one tenth the attack radius) around the adversarial example propagating this box through the network. In our preliminary testing we found that for CIFAR-10 8/255 we get a baseline performance of $0.0054 < 0.0062$ and a certified performance of $0.0151 < 0.0153$. Here

we find that using these smaller bounding boxes increases our performance on both metrics with an especially large increase in the baseline performance (inline with observations from SABR). Our paper focuses on introducing the idea of support sets for certified VAE training and is orthogonal to developments in standard certified training. We leave additional performance gains achievable by combing CIVET with other methods for future work.

# H ABLATION FIGURE

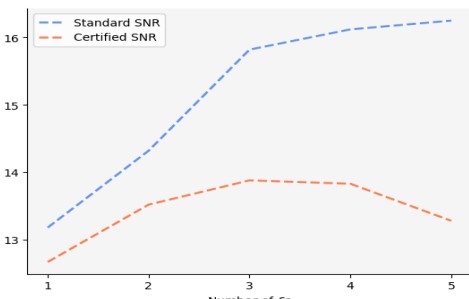 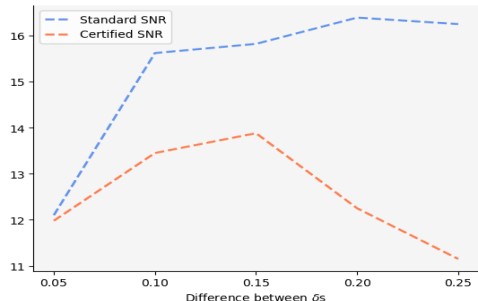

Figure 3: Standard and Certified SNR while varying $\mathcal{D}$. We consider sets with $\delta_n = 0.05$ and $\delta_i = \delta_{i+1} + \eta$ (let $n = |\mathcal{D}|$). On the left, we vary the size of $\mathcal{D}$ between 1 and 5 while fixing $\eta = 0.15$. On the right, we fix $|\mathcal{D}| = 3$ and vary $\eta$ between 0.05 and 0.25.

