# OpenReview forum: "Support is All You Need for Certified VAE Training"
_ICLR.cc/2025/Conference — ICLR 2025 Poster_

### Official Review · Reviewer_BKMJ · 2024-10-22

**Soundness:** 3
**Presentation:** 3
**Contribution:** 2
**Rating:** 5
**Confidence:** 3

**Summary:**

Variational Autoencoders (VAEs) have been shown to be susceptible to adversarial attacks, but can be trained to become more empirically robust to such attacks using adversarial training which employs an underapproximation of the worst-case loss. However, training certifiably robust VAEs is difficult due to the fact that for a given input, the encoder usually generates the parameters of a distribution in the latent space instead of a concrete vector. The authors show that by finding a support set for the latent space distributions, one can derive an upper bound of the worst case loss using standard interval propagation methods. The authors introduce a method for finding support sets in a restricted space which result in tight upper bounds. These sets can then be used to compute a concrete upper bound of the loss which forms part of the CIVET training algorithm presented in the paper. CIVET is evaluated on a VAE for a wireless networking problem and computer vision problems where it outperforms standard and adversarial training in terms of certified performance. When compared to Lipschitz VAEs as a baseline for certifiably robust VAEs, it achieves comparable certified performance while enjoying an improved standard performance.

**Strengths:**

- The presented method is sound and theoretical results are accompanied by thorough proofs
- CIVET achieves significantly higher certified performance than standard and PGD training
- The method is novel and allows for certified training of standard VAE architectures

**Weaknesses:**

The experimental results are somewhat underwhelming. Table 1 shows that CIVET achieves lower standard performance than standard and PGD training but higher certified performance. The decreased standard accuracy is to be expected given the fact that certified training has an implicit regularisation effect [1]. The increased certified accuracy here is also expected given that the loss in standard and adversarial training does not favour certified robustness in any way. These are very weak baselines in terms of certified accuracy and it is not surprising that CIVET would outperform them in terms of certified performance. Section 4.4 also only compares CIVET to those baselines. The more interesting baseline are the Lipschitz VAEs since these are designed to be certifiably robust. I find the results in section 4.2 difficult to process in their current form, but they are not that convincing in my opinion (slightly better baseline performance and similar certified performance).

Generally, the experimental evaluation is somewhat underdeveloped since only two experiments which compare the approach to a baseline for certifiably robust VAEs are conducted ($\epsilon=0.1$ and $\epsilon=0.3$ on MNIST and one model). Additional experiments on different epsilon sizes/datasets or experiments varying the latent space dimensions/network depths should be run to provide a more thorough comparison.

The authors should also move at least some of the information on the runtimes to the main part of the paper. Training VAEs using CIVET takes ~10x as long as standard training. This is extremely expensive, even for a certified training method, yet is not discussed in the paper at all. A proper runtime comparison between the algorithms, including the time needed for training Lipschitz VAEs, should be added to make a comparison possible, for example using a table.

Due to the above points I believe that this paper would significantly benefit from another revision cycle, allowing for the extension of the experiments section so a proper comparison between the work and other methods is possible.

## Minor points
- Line 490: CIVET maintains its lead against real-world attacks --> The decreased standard accuracy compared to other training methods could be mentioned here
- The results in Section 4.2 should be summarised in a table, at the moment they are very hard to process since they're only given in the text

## Literature
[1] Mao, Y., Mueller, M.N., Fischer, M. & Vechev, M. (2023) Understanding Certified Training with Interval Bound Propagation.


## Editorial
- Line 93: an approximated posterior distributions --> an approximated posterior **distribution**
- Line 102: The abbreviation "SNR" is used before it is explained
- Line 131: Then refines the network --> **The method** then refines the network
- Line 185: the worst case error (...) --> the worst case error **is**
- Line 192: possible output distribution --> possible output distribution**s**
- Line 203f: finding appropriate subset --> finding **an** appropriate subset
- Line 203: compute non-trivial upper bound --> compute **a** non-trivial upper bound
- Line 259: So only picking support --> So only picking **a** support
- Line 264 (caption of Figure 2): The caption defines $S_\delta = [\mu_{lb} + \zeta, \mu_{ub} + \zeta]$, I think this should be $S_\delta = [\mu_{lb} - \zeta, \mu_{ub} + \zeta]$ (wrong sign for the lower bound of the interval)
- Line 271: Let's first consider 1d case --> Let's first consider **the** 1d case
- Line 278f: the interval $[l, u]$ always includes the interval $[\mu_{lb} + \mu_{ub}]$: Not sure what is meant here, should this be $[\mu_{lb}, \mu_{ub}]$
- Line 318: gives us a way to find support set --> gives us a way to find **a** support set
- Line 327: we weight them based on --> we **weigh** them based on
- Line 365: implemented CIVET in in PyTorch --> implemented CIVET **in** PyTorch (remove one "in")
- Line 459: A network is $f: [...]$ is Lipschitz continuous --> A network $f: [...]$ is Lipschitz continuous (remove one "is")
- Line 466: Lipchitz --> Lip**s**chitz
- Line 480: We leave further study of $\mathcal{D}$ selectin --> We leave further study of $\mathcal{D}$ selecti**o**n
- Line 511: using IBP to obtain certfied --> using IBP to obtain cert**i**fied

**Questions:**

- What are the main reasons for the long runtime of the CIVET algorithms compared to other algorithms? Is there any way of possibly accelerating the proposed method?
- Could the authors provide the training times for the Lipschitz VAEs?
- Do the authors have any additional experimental results which compare their method to other method for obtaining certifiably robust VAEs?

---

### Official Review · Reviewer_Wi5q · 2024-10-30

**Soundness:** 2
**Presentation:** 2
**Contribution:** 3
**Rating:** 6
**Confidence:** 4

**Summary:**

The submission presents CIVET, a novel algorithm to perform certified training of VAEs.
Differently from a previous work that does this by building a low-Lipschitz network, the authors devise a way to do certified training using network relaxations (bounding techniques).
In order to be able to do so, the authors present a way to find a "support set": a set of allowed latent space perturbations that captures a probability mass (under the considered adversarial model) at least as large as the probability with which the networks needs to be verified.
Once this is in place, the computation of an upper bound to the worst-case loss to be employed at training time mirrors the procedure employed for standard feedforward networks.
Experimental results on vision and on a wireless communications dataset are presented.

**Strengths:**

The idea of computing a "support set" associated with the desired verification probability to then be able to use techniques from standard feedforward networks to upper bound the worst-case loss is novel to the best of my knowledge. I think this is an elegant solution to the VAE certified training problem.
The technical contributions (how to determine the support set) are also non-trivial.
The wireless communication benchmark provides a very solid motivation to the approach, providing a compelling non-vision benchmark (not common in certified training).

**Weaknesses:**

I think the paper would definitely benefit from some polishing on the writing side, and from a more detailed and reproducible experimental section.

On the writing side, I think some examples (perhaps in an appendix) would definitely aid the reader in section 3. I found Figures 1 and 2 to be helpful in this regard, but not very readable when printed (especially if in black and white).
Sometimes concepts are introduced before being explained: for instance, in line 325 the fact that there is a list of deltas. It is not until the experimental section that it is clarified that these are given to the algorithm.
Some design choices (such as the weighting of the for the various losses associated to each delta) are not well-justified.
Section 4.2 would definitely benefit from a table view of the presented results.
It would also be nice to have an appendix providing more details on the FIRE dataset (why a VAE is used there, what is the rationale, etc).

Furthermore, plenty of details are omitted from the experimental section (and the associated appendix). Given that the code is omitted (it is promised by the authors, yet not provided), at the very least one should get a complete description of the experiments. The training schedule and hyper-parameters are, for instance, unclear. Same goes for the (rather important) details of the adversarial training baseline. Additional examples in the questions below.

Concerning the experiments themselves, I found some of the results/claims to be somewhat surprising, and worthy of further discussion. In certified training for feedforward networks, IBP typically incurs a very large standard performance cost. The authors, nevertheless, say that "*CIVET obtains significantly better certified performance without sacrificing much baseline performance*". It does not seem to be the case for more challenging datasets at larger perturbations (CIFAR-10, 8/255), where performance is significantly reduced.
Furthermore, the results from the Lipschitz VAEs are provided only for MNIST. However, this is quite an important baseline. One could for instance extrapolate that Lipschitz VAEs perform better at larger epsilons. Would it be the case on CIFAR-10?
Finally, I was also surprised to see that the only attack from previous work being tested as a UAE physical attack on the FIRE benchmark. Could the weakness of the attack (imposed, for instance, by the universality constraint) be behind the superior empirical robustness of CIVET compared to PGD? In the certified training literature, it is typically assumed that certified training comes at an empirical robustness cost, provided the adversarial training baseline is strong enough.

**Questions:**

- Can you please discuss the similarities and differences between your approach (mainly, the idea to use a support set), and the techniques to perform BNN verification and IBP training introduced by (Wicker et al., 2020) and (Wicker et al., 2021)? I believe they also rely on bounding boxes (on the weights perturbations) which then allow to rely on techniques from standard feedforward networks.
- A series of certified techniques, including the cited (Balunovic and Vechev, 2020), (Mueller et al., 2022), (Mao et al., 2023) and [1,2,3], proposes to use a mix of adversarial training and network bounding to improve both certified accuracy and standard performance. It would be nice to acknowledge these developments in the related work section. It would be even better if the authors could test some of these ideas on CIVET: would such techniques improve performance in this context too?
- Could you please elaborate on the choice of the specific weighting for the various losses associated to each delta? Why not something like $\sum_i (1 - \delta_i) \mathcal{L}_i$? How was the $\mathcal{D}$ set obtained in the experiments? Are these all hyper-parameters? Looking at Figure 3, it would seem like some tuning was carried out.
- What is the $\delta$ employed for the certification in Table 1, section 4.2 and Figure 3?
- The output of a verification query is binary ({True, False}). What does certified performance mean, in this context, then? How is the provided number obtained from the framework from (Berrada et al., 2021)?
- Could it be possible to compare against Lipschitz VAEs on CIFAR-10?
- Can Table 1 include a comparison in terms of empirical robustness to previously-published VAE attacks? Many are cited in (Barrett et al, 2022).
------

[1] Xiao et al.. Training for faster adversarial robustness verification via inducing relu stability. ICLR, 2019.

[2] Fan and Li. Adversarial training and provable robustness: A tale of two objectives. AAAI, 2021.

[3] De Palma et al.. Expressive losses for verified robustness via convex combinations. ICLR, 2024.

---

### Official Review · Reviewer_7PDy · 2024-11-06

**Soundness:** 3
**Presentation:** 3
**Contribution:** 3
**Rating:** 6
**Confidence:** 3

**Summary:**

This paper explores the certification and related questions of VAEs. VAEs are a type of neural network with outputs that contain probability distributions; therefore, standard certifiable robust training does not work for them. The authors first introduce a theoretical framework called support sets to bound the worst-case loss, and then provide an algorithm to select a set of supports.  Based on this, the authors introduce the concept of CIVET loss and design a training algorithm using CIVET loss. They show that CIVET training performs better compared to standard and PGD training methods on various datasets and better than Lipschitz VAEs.

**Strengths:**

1. This paper has a solid theoretical foundation and provides a results table to show its good performance. In Table 1, compared to standard and PGD training, CIVET training outperforms in all cases in the Certified column with a significant advantage. Compared to Lipschitz VAEs, CIVET training obtains better baseline performance with comparable certified performance.

**Weaknesses:**

1. Also in Table 1, in the Baseline column, CIVET training is never the best and is sometimes significantly worse, while PGD and standard trainings usually are close in performance.

2. Compared to Lipschitz VAEs, the improvement in baseline performance (3.8e-3 vs 4.9e-3) does not look amazing, and for certified performance, CIVET is not better than Lipschitz VAEs.

Considering these two points, although CIVET shows good performance, it is hard to consider it as SOTA.

**Questions:**

1. Did the authors compare CIVET to more training methods?

2. Between baseline and certified performance, does this paper consider one more important than the other?

3. Is 3.8e-3 vs 4.9e-3 indeed a big improvement?

---

### Meta-Review · Area_Chair_tbLg · 2024-12-20

**Metareview:**

The submission presents CIVET, a novel algorithm to perform certified training of VAEs.

+ The proposed approach appears technically sound.
+ The problem studied is interesting.

- The paper lacks clarity.
- Empirical evaluation is limited.

**Additional Comments On Reviewer Discussion:**

Some technical issues were raised in the reviews, including
- choice of the specific weighting for the various losses
- comparison to more training methods
- significance of reported improvements

These were partially clarified in the rebuttal. However, even after the discussion phase, the strength of the baseline is unclear. Specifically, whether the baseline was tuned appropriately or not. This needs to be better explained in subsequent versions of the paper.

---

### Decision · Program_Chairs · 2025-01-22

Accept (Poster)